# Environmental Stressors and the PINE Network: Can Physical Environmental Stressors Drive Long-Term Physical and Mental Health Risks?

**DOI:** 10.3390/ijerph192013226

**Published:** 2022-10-14

**Authors:** Nicolas J. C. Stapelberg, Grace Branjerdporn, Sam Adhikary, Susannah Johnson, Kevin Ashton, John Headrick

**Affiliations:** 1Gold Coast Hospital and Health Service, Gold Coast, QLD 4215, Australia; 2Faculty of Health Sciences and Medicine, Bond University, Gold Coast, QLD 4226, Australia; 3Mater Young Adult Health Centre, Mater Hospital, Brisbane, QID 4101, Australia; 4School of Medical Science, Griffith University, Gold Coast, QID 4215, Australia

**Keywords:** psycho-immune-neuroendocrine network, chronic illness risk network, biological networks, metabolomic pathways, mental health, non-communicable disease, biopsychosocial model, syndemic, particulate matter

## Abstract

Both psychosocial and physical environmental stressors have been linked to chronic mental health and chronic medical conditions. The psycho-immune-neuroendocrine (PINE) network details metabolomic pathways which are responsive to varied stressors and link chronic medical conditions with mental disorders, such as major depressive disorder via a network of pathophysiological pathways. The primary objective of this review is to explore evidence of relationships between airborne particulate matter (PM, as a concrete example of a physical environmental stressor), the PINE network and chronic non-communicable diseases (NCDs), including mental health sequelae, with a view to supporting the assertion that physical environmental stressors (not only psychosocial stressors) disrupt the PINE network, leading to NCDs. Biological links have been established between PM exposure, key sub-networks of the PINE model and mental health sequelae, suggesting that in theory, long-term mental health impacts of PM exposure may exist, driven by the disruption of these biological networks. This disruption could trans-generationally influence health; however, long-term studies and information on chronic outcomes following acute exposure event are still lacking, limiting what is currently known beyond the acute exposure and all-cause mortality. More empirical evidence is needed, especially to link long-term mental health sequelae to PM exposure, arising from PINE pathophysiology. Relationships between physical and psychosocial stressors, and especially the concept of such stressors acting together to impact on PINE network function, leading to linked NCDs, evokes the concept of syndemics, and these are discussed in the context of the PINE network.

## 1. Introduction

In 1977, Engel proposed the biopsychosocial model to describe a holistic framework for the determinants of disease. Engel proposed that “… a medical model must also take into account the patient, the social context in which he lives, and the complementary system devised by society to deal with the disruptive effects of illness…” [1]. The biopsychosocial model considers biological, psychological, as well as social and environmental factors as determinants for disease [2]. This model was embraced as a framework for understanding mental illness in particular; however, it may also act as a useful framework for understanding many chronic non-communicable diseases (NCDs). This framework is possibly also an early attempt at understanding disease within a systems context.

Since the introduction of the psycho-immune-neuroendocrine (PINE) network model [3], it was asserted that stress (an environmental factor) acted as a potent driver for the emergence of pathophysiology which can lead to one or multiple mental health and medical conditions [4,5]. These multiple (and sometimes multimorbid) conditions were described in terms of a chronic illness risk network (CIRN) [5], with non-communicable diseases (NCDs) including atherosclerosis and coronary heart disease (CHD), type 2 diabetes mellitus (T2DM), cerebrovascular accident (CVA) and vascular cognitive impairment (VCI), together with mental disorders such as major depressive disorder (MDD).

While psychosocial stressors have long been implicated in the onset of CIRN conditions, we assert that physical environmental stressors are similarly responsible for PINE pathophysiological change. We propose that environmental factors act both as risk factors and modifiers for PINE network dysfunction and progression to NCDs, following a stress diathesis model originally described by [6] (Bleuler, 1963), and explored briefly in relation to the PINE model [7]. The syndemic framework was developed to understand interacting diseases promoted by social, economic, environmental and political features of a population. The hallmark of a syndemic is the presence of two or more disease states that adversely influence each other and enhance vulnerability, and which are exaggerated by socioeconomic factors. One potent example of a physical environmental stressor is airborne particulate matter (PM), which can impact the PINE network, potentially leading to NCDs. PM is classified based on particle size, and there are three main classifications [8]. PM10 particles, also known as course particles, are particles ≤10 µm in diameter commonly released via direct, rather than secondary, environmental stimulation such as dust storms or construction work [9]. PM2.5 particles are known as fine particles with a diameter ≤2.5 µm, and due to their small size, travel further and accumulate greater than PM10, and penetrate deeper into human tissues [10]. PM2.5 originate primarily from fossil fuel combustion. Finally, PM0.1, known as ultra-fine particles, are particles ≤0.1 µm in diameter [11]. These particles are commonly produced by combustion processes and are formed by the coalescence of ions.

We review the evidence that chronic (and even relatively brief) exposure to PM induces a biological stress response that detrimentally modifies integrated physiological pathways in the PINE network, giving rise to chronic medical conditions [5] via disruption of specific PINE subsystems such as immune, endocrine (especially the HPA axis) and autonomic nervous systems [3], together with other physiological systems such as adipokine signalling, gut permeability and the microbiome [7]. The medical conditions detailed extend substantially beyond respiratory conditions and potentially involve intergenerational impacts.

Importantly, there is less evidence for biologically driven mental health conditions linked to PM. However, we argue for the theoretical existence of these in the discussion, along with recommendations to pursue evidence to support this claim. An overview of the potential relationships between diathesis, environmental stressors (e.g., PM), the PINE network, health and potential mental health sequelae is shown in Figure 1. The primary objective of this review is to explore evidence of relationships between airborne PM, the PINE network and chronic NCDs.

## 2. The PINE Network and Non-Communicable Diseases: PM as a Chronic Stressor

Since the identification of general adaptation syndrome by Selye in 1936 [12], describing the impact of chronic stress on physiological systems, and the later establishment of allostatic theory by McEwen and colleagues [13,14,15], there has been a recognition that chronic stress influences multiple interlinked systems, producing pathophysiological changes that lead to chronic disease. Viewing these interlinked pathways as a network using a systems biology approach, the PINE model provides insights as to how behaviour of the entire network can explain transition from health to MDD and other NCDs [3]. This work was expanded to illustrate dynamic behaviour of the PINE network with chronic stress, including feedback loop transitions and a critical transition to disease [4,5]. The concept of a CIRN then proposed that the increased reciprocal risk between several major NCDs may reflect shared causative pathways within the PINE network; notably immune, autonomic and endocrine dysregulation [5].

Chronic stress acts on the autonomic nervous system (ANS) and HPA axis [16,17], resulting in sympathovagal imbalance, parasympathetic withdrawal [18,19] and sympathetic overdrive [19]. The long-term stimulation of the HPA axis results in cortisol resistance [16,19,20,21,22]. The negative regulatory feedback loops in the ANS and HPA axis that normally maintain homeostasis are therefore disrupted by chronic stress. Both vagal withdrawal and glucocorticoid resistance may ultimately push some negative feedback loops through a transition to positive feedback, driving a system-wide transition to disease [4]. Via neural and endocrine modulation of acute inflammation [23,24,25], immune function is intimately linked to this chronic stress response, increasing circulating inflammatory markers [3,4]. This systemic pro-inflammatory state promotes neuroinflammation, microglial proliferation and astrocyte loss [26,27,28], kynurenine pathway (KP) activation, tryptophan depletion and reduced CNS serotonin production [29]. Overactivity of the KP increases neurotoxic metabolites, inhibiting neurotrophic signalling and neuronal maintenance to further suppress astrocyte numbers [29,30,31]. Metabolite effects on hippocampal neurogenesis [32,33] in turn may influence learning [34] and memory [35]. These different CNS changes have all been linked to MDD: inflammation is strongly linked to MDD [36] and inflammatory mediators such as IL-1 drive sickness behaviours [37,38,39] that are identical to the neurovegetative symptoms and anhedonia of MDD [3,5,40]; reduced CNS serotonin and brain-derived neurotrophic factor (BDNF) are both linked to MDD [41], and shifts in neurogenesis and BDNF signalling may mediate cognitive changes in MDD [5,42].

Other systems constitute elements of the PINE network, notably the gut and gut microbiome. Dietary factors and inflammation affect gut motility [43] and permeability [44], facilitating the entry of bacteria and associated lipopolysaccharides (LPSs), inducing systemic pro-inflammatory cytokine release in a positive feedback loop [38,45,46]. Recent evidence indicates PM exposure also induces gut dysbiosis in animals and humans, promoting inflammation and metabolic disease [47,48,49]. The pro-inflammatory state arising with chronic stress increases leptin and reduces ghrelin secretion [50], leading to hypothalamic leptin resistance [51,52]. This central leptin resistance further increases circulating leptin levels, which in a positive feedback manner further enhances inflammatory cytokine release [53,54,55,56,57]. Through this array of changes, spanning multiple biological processes and pathways, long-term stress may give rise to a variety of chronic disorders [5]. We assert that PM may serve as an environmental trigger of PINE network disruption and then downstream, stress-related disease processes.

## 3. Air Pollution Induces a Stress Response and Disrupts the PINE Network

There is increasing evidence of a relationship between air pollutants and a stress response, evidenced by changes in biological markers characteristic of chronic stress and we review evidence of air pollutant impact on key PINE subsystems: Immune, endocrine, autonomic and the gut and gut microbiome.

### 3.1. Pro-Inflammatory Response

Inhaled PM triggers pulmonary protein leakage and inflammation in a size-, dose- and solubility-dependent manner [58,59]. The persistence of inhaled particulates may prolong these effects. Exposure to PM influences multiple organ systems indirectly via the production of inflammatory factors and triggering of nervous reflexes, and directly via access to the circulation where they disrupt cellular, tissue and organ function to modulate autonomic nervous system (ANS) activity [60,61,62,63]. The impact of PM on pro-inflammatory responses are associated with the size of the particles. Due to their size, PM10 are restricted to the pulmonary system, where they locally increase reactive oxygen species (ROS) and inflammatory cytokines [64]. PM2.5 are associated with elevated inflammation and oxidative stress, together with changes to ANS activity [10]. Exposure to PM2.5 elevates inflammatory markers including CRP, TNF-α, PGE2 and Il-1α together with ET-1 [65,66]. PM0.1 are also heavily implicated in airway diseases, but have also been shown to be associated with inflammation in the cardiovascular and central nervous system [11] (Nelin et al., 2012).

Chronic inflammation is linked to proximity to roadways, where there are increased PM levels [67]. Inhalation of poorly soluble pollutants such as ozone, NO_2_ and phosgene also induces pulmonary inflammation [68,69], with changes focused within the lower respiratory tract [70,71]. Conversely, water soluble sulphur dioxide, chlorine and ammonia may predominantly influence the nasopharynx [70,72,73,74]. Volatile organic chemicals in vehicle and other pollutant sources are also detrimental to cardiovascular health [75], and lead to immunosuppression [76] and Alzheimer’s disease [77]. For example, acute acrolein exposure induces dyslipidaemia [78] and vascular dysfunction [79,80], while chronic exposure promotes atherosclerosis and lesion rupture [80,81] and impairs protective anti-infarct signalling [82].

Microparticulate exposure additionally triggers adipose tissue production/release of cytokines, contributing to systemic inflammation and hepatic insulin-resistance [71]. There is also evidence of immune cell mobilisation from bone marrow in response to PM exposure [83,84,85]. Despite these clear pro-inflammatory changes, mixed effects of air pollution on circulating cytokines are still reported, ranging from increases in humans and animals [86,87,88] to no apparent change [89,90,91].

These different peripheral changes and inflammatory processes contribute to a neurogenic inflammation with air pollution exposure [92,93]. Activation of neuronal and epithelial transient receptor potential proteins (TRPs) by pollutant products triggers a tachykinin (e.g., substance P, neurokinin A) release [94,95,96], whilst capsaicin-sensitive TRPV1 channels evoke a neuronal tachykinin release [97], and ion channel receptors trigger a neuronal neurokinin release [98,99]. This induces neurokinin-receptor-dependent pulmonary inflammation [100,101], with pulmonary injury and cytokine production spilling over into the systemic circulation, contributing to metabolic changes and propagating vascular and CNS inflammation [102,103]. Neuroinflammation, in turn, contributes to mental health issues such as MDD and other NCDs via perturbation of the PINE network [3,5].

Inflammatory mediators produced with pollutant exposure may activate glial cells in the CNS [104]. Interestingly, cytokine-mediated NFκB signalling in the hypothalamus appears to enhance particulate-induced systemic inflammation and metabolic changes [71], supporting positive feedback augmentation of inflammation via the CNS. Central stress-axis regions such as the paraventricular nucleus (PVN) of the hypothalamus may also be activated via sensory irritant activation of the trigeminal and vagal nerves with pulmonary oxidative stress/irritants, stimulating sympathetic and HPA-axis activities and hormone release.

These central effects of air pollutants are influenced by, and in turn modulate, systemic metabolic changes. Experimental studies in rodents identify links between long-term particulate exposure and pollution-triggered inflammation with adipose and hepatic inflammation, hepatic ER stress, glucose intolerance and insulin-resistance [103,105]. These observations are consistent with links between chronic particulate exposure and insulin-resistance, hyperglycaemia and diabetes in humans [106]. Studies of acute exposure to gaseous pollutants reveal that lipid oxidation by-products are elevated, promoting skeletal muscle insulin-resistance [107].

### 3.2. HPA-Axis Activation and Dysregulation

Though fewer studies implicate stress hormones and their receptors in environmental disease susceptibilities [108], there is substantial evidence of stress-axis modulation in response to air pollutants [58]. Ozone has been extensively studied as a specific airborne toxin; however, any air pollutant that interacts with biological components of the airway and triggers neural responses can stimulate a HPA-mediated stress response [58]. Virgolini and colleagues [109,110] demonstrated permanent alteration in HPA-axis function with lead exposure, and heavy metals can form a significant component of air pollution. Heavy metal concentration can also be found in bush fires, with fires in Lithuania being linked to increases in heavy metals such as copper, lead and zinc contaminating river water [111], while ash from Californian wildfires in 2007 was shown to contain substantial levels of arsenic, cadmium, copper and lead [112].

It has been argued that the environmental stress of air pollution and psychological stress may act synergistically in disrupting health. Clougherty and Kubzansky [113] provide a compelling case for air pollution and stress potentiating respiratory disease onset and severity, with a lucid discussion of the central role of the HPA-axis, while Olvera Alvarez et al. [114] propose that early life stress and air pollution act synergistically to increase the risk of chronic diseases, including MDD, CHD, T2DM and lung and brain cancer [115]. These authors suggest early life stress results in long-term modification of HPA-axis function (e.g., via NR3C1 glucocorticoid receptor gene methylation), rendering the HPA axis susceptible to dysfunction with environmental stressors such as air pollution [114]. Similarly, [3,5] suggest that early developmental diathesis contributes to later disease risk via the PINE network, a ramification of which is that people experiencing early life stress and subsequently exposed to events such as bushfires or elevated levels of PM may be at greater risk of multiple chronic illnesses, including MDD, CHD or T2DM.

### 3.3. Autonomic Dysregulation

Air pollution can be linked to these chronic disorders via inflammatory, chemical and ischemic influences on the ANS [116]. To measure cardiac autonomic control, heart rate variability (HRV) is a well-recognised non-invasive and quantitative marker reflecting rhythmic activity of the sinus node, which is analysed in time, frequency or non-linear domains [117]. While HRV measures generally reflect sympathovagal balance, observed HRV may be primarily driven by vagal activity [118,119,120], with reduced HRV linked to vagal withdrawal [18,19]). Reduced HRV is associated with poor health outcomes and is linked to conditions such as MDD and CHD [121] and pathophysiological changes in the PINE network (discussed above).

Air pollution, especially PM2.5, is associated with decreased HRV in different populations, including healthy young adults (particularly when simultaneously exposed to amplified noises) [122], healthy adult boilermakers [123], healthy senior adults [124,125,126], and those with cardiovascular diseases [127,128,129] also identified a decrease in HRV with PM exposure in elderly people, who were non-smokers and had no serious medical conditions. Additional smaller studies compared personal exposure to PM with HRV in healthy and diseased individuals [130,131,132]. For individuals, particularly the elderly, with certain underlying respiratory conditions, PM exposure can have up to a 4-fold increase in autonomic instability compared healthy young individuals [132]. Alongside reducing HRV, there is also limited evidence that PM increases coagulation [131]. Although limited by small sample sizes, these studies add to a body of evidence linking PM exposure, autonomic dysregulation and inflammation and indicating that pre-existing diseases can increase vulnerability to autonomic imbalance.

There is noted heterogeneity within study findings. [130] (2003) report that for (a limited cohort) patients with stable, severe CHD, CO but not PM exposure may briefly modify autonomic control. Furthermore, studies of older people presenting with chronic obstructive pulmonary disease (COPD), found no associations between HRV and PM2.5 [133,134], potentially reflecting differences in systemic inflammation compared to other subgroups [135], including a baseline lower HRV in individuals with COPD [136,137]. Taken together, these results suggest that the acute cardiovascular toxicity of PM alters the autonomic control of the heart depending on the underlying health status of the individual.

## 4. Evidence PM Exposure Promotes Chronic Non-Communicable Diseases

The acute or direct health effects of elevated background PM exposure and following significant events, such as bush fires, has been relatively well documented [138]. For instance, exposure to smoke is associated with increased hospitalisations [139], respiratory morbidity such as asthma and COPD [140], stroke [141], cardiovascular disease [142], poorer birth outcomes [143] and premature deaths [144]. Analysis of acute smoke exposure and premature deaths in Sydney from 1994 to 2007 reveals a 5% increase in mortality during bushfires [145]. In terms of the acute effects of recent fires in eastern and southern Australia, [146] estimated ~420 excess deaths from this smoke exposure, together with 3151 additional cardio-respiratory related hospitalisations and 1305 emergency department attendances for asthma.

In addition to immediate or early effects, air pollution has been linked to later development of chronic diseases and emerging evidence supports epigenetic promotion of such disease risk across generations. The WHO estimates that >80% of those individuals in urban areas are exposed to air pollution levels exceeding guideline limits [147], with pollution disproportionately impacting those in less economically developed populations [148], mirroring the demographics for major NCDs. Inhaled pollutants impact organ systems beyond the lungs (Kurt et al., 2016), including the heart and blood vessels [149,150,151], liver [152], kidneys [153,154] and as detailed above, the CNS [155]. Air pollution also affects developmental programming [156].

Beyond predictable associations with respiratory disorders [157], air pollution is linked to NCDs including steatohepatitis [158], diabetes [106], neurodegenerative diseases [159] and cancers [115,160]. Cardiovascular disorders, such as hypertension [161] and CHD [162,163], are particularly strongly associated with air pollution. Even brief exposure is linked to AMI, stroke, arrhythmias, worsening of heart failure and hypertension [164,165,166,167], while chronic exposure accelerates atherosclerosis, impacts blood pressure control, thrombosis, endothelial function, insulin sensitivity [164,166] and increases the risk of hypertension, Long-term effects of ambient PM2. 5 on hypertension and blood pressure and attributable risk among older Chinese adults [167,168]. The WHO estimates that air pollution contributes to ~7 million premature deaths globally per annum, with cardiovascular impacts, for example, rivalling the most widely studied and powerful drivers of smoking, hypertension and inactivity [165]. More recent analysis ascribes 9 million premature deaths to pollution in 2015 (16% of global deaths), and welfare costs approaching USD 5 trillion annually [169]. There appear to be no safe levels of air pollution in terms of increased mortality [170].

Exposure to PM2.5, predominantly generated from fossil fuel combustion and bushfires, is linked to premature death due to cancer, respiratory, metabolic and cardiovascular diseases [138]. As with air pollution more broadly, those with existing conditions are particularly sensitive to PM exposure [166], exacerbating acute cardiovascular events and promoting chronic CVD. While the relatively large size of PM10 particles (predominantly pollen, dust and construction or agriculture by-products) limits penetration to the upper respiratory tract, PM2.5 and smaller reach the alveoli which may underlie greater impacts of PM2.5 on multiple organ systems.

One body of evidence linking airborne PM to NCDs comes from studies of air pollution exposure in socioeconomically disadvantaged people. The markedly increased NCD risks in minority and low-socioeconomic-standing (SES) groups may involve disproportionate exposures to pollutants. For example, diabetes is linked to increased pollutant exposure in high-risk groups in the USA [171,172,173]. Proximity to major roadways, strongly linked to low SES, is associated with increased carotid-intima thickness [174], abdominal adiposity [175], hypertension [77], the risk of AMI [176] sudden cardiac death [177], mortality due to acute heart failure [178] and stroke [179]. Concentration of chemical plants and waste sites in low SES communities also contributes to increased exposure and thus disease risk [180]. The health impacts of pollution are also influenced by SES, dietary and other factors. For example, a lack of association between PM exposure and CVD in a study of male health professionals has been attributed to a higher SES and healthier lifestyle of this sub-population [181]. In summary, PM is linked to NCDs in both short-term and longer-term timeframes. We have detailed how PM impacts PINE subsystems and have shown that PM is linked to not only respiratory disease, but NCDs that have previously been linked to PINE network pathophysiology [5].

Additionally, emerging research suggests that air pollution and PMs induce epigenetic changes that promote chronic disease development [182] and influence disease risk in offspring. For example, the epigenetic effects of PM may increase susceptibility to, and progression of, lung cancer for the next generation [183]. Altered methylation due to air pollution is also linked to other respiratory conditions [184], and increases the risk of cancer more broadly [185]. Toxic air pollutant mixtures alter blood DNA methylation levels for mitogen-activated protein kinase pathway components [184,186], and functional changes to blood-based measures of protein expression [160,187,188]. Such DNA methylation changes with air pollution exposure are evident across the lifespan [189], affecting both children [190] and the elderly [191]. In pregnancy, air pollution has the capacity for developmental reprogramming of the epigenome, with exposure linked to differential shifts in methylation in early and later pregnancy [192,193]. These epigenetic influences can serve to entrench chronic disease risk in disadvantaged populations disproportionately exposed to high air pollutant levels.

## 5. Mental Health Sequelae of Air Pollution

While a body of literature from the 1960s onwards documents direct and indirect psychological effects of individual components of smoke or pollution (PM, photochemical oxidants, nitrogen oxides, sulphur oxides, carbon monoxide and ozone) and overall air pollution load [194], there is limited evidence linking air pollutants to chronic mental illnesses such as major depression, and we elaborate on this issue in the discussion.

Higher general pollution levels increase physiological and psychological stress [113], while individual components exert specific effects that may vary considerably [195]. Carbon monoxide (CO) reduces vigilance and ability to perform repetitive and monotonous tasks [196], whereas ozone decreases nocturnal and peripheral vision [197]. The presence of a moderately noxious odour has also been shown to increase aggressive tendencies, compared to a mild or severely noxious odour (which produce either no change or an escape response) [198]. An early study [199] reviewed rates of Emergency Department presentations and admissions for psychiatric illness against daily pollution levels, reporting a statistically significant increase in psychiatric presentations with higher daily CO levels, with non-statistically significant correlations between nitrogen dioxide (NO_2_) and admissions for alcohol dependence and organic brain syndromes. They also noted a negative correlation between NO_2_ and admission rates for unknown and non-psychotic diagnoses [199], which was discussed in relation to the anaesthetic/analgesic properties of NO potentially counteracting the irritant effects of NO_2_.

## 6. Discussion

Exposure to PM has many direct health effects, evidenced by short-term increased presentations with respiratory, cardiovascular and other health issues. However, evidence is also presented for a longer-term greater incidence of chronic medical conditions arising after exposure to air pollution. While there are direct mechanisms which account for some of these increased incidences, many of these diseases are also influenced by perturbation of the PINE network. We have explored evidence of more indirect and long-term immune, HPA-axis and autonomic changes with exposure to airborne particulate matter, contributing to biological changes in the PINE network, that are consistent with the effects of chronic stress. These changes may facilitate critical transition of the PINE network to a pathophysiological state, leading to chronic NCDs such as CHD, T2DM, stroke or cancer. Intriguing is the question of whether PINE network disruption, driven by elevated levels of environmental PM, could equally result in mental health conditions such as MDD in the longer term. We have demonstrated that air pollution and airborne PM bring about biological PINE network disruption and assert that this may occur by PM acting as a biological stressor, with PINE effects similar to psychosocial stressors, as illustrated in Figure 1. This implies that there is a potential risk of physical environmental stressors precipitating mental illness, not just chronic medical illness, by acting on the PINE network.

### 6.1. Evidence for Long-Term Health and Mental Health Sequelae from Exposure to Air Pollution

The hypothesis postulating downstream mental health sequelae following exposure to chronic (e.g., air pollution) and severe acute events (e.g., bush fires), should be supported with evidence of increased mental illness, not explainable by other causes, over a longer-term timeframe, but it is limited. There is some evidence supporting increased incidence of developmental [200] and neurodegenerative [201] conditions as sequelae of chronic exposure to high level air pollution. In terms of mental health sequelae, a meta-analysis [202] found a possible correlation between rates of depression and suicide with long-term PM2.5 exposure. There were also significant confounders that could not be adjusted for including stress related disease [203].

There is however little evidence for long-term effects of acute exposure. [204] reported consistent and increased anxiety-like behaviours and depressive features in mice exposed to smoke, in association with increased pro-inflammatory cytokines, decreased myelination and hippocampal astrogenesis and microgliosis (consistent with changes evident in experimental disease models and MDD) as a consequence of this exposure. Exposure to bushfires results in long-term mental health sequelae such as MDD, anxiety disorders, post-traumatic stress disorder (PTSD), somatisation and suicidality [205,206,207], but these sequelae are likely be related to trauma from the bushfire events themselves.

### 6.2. Risk Timeframes and Amount of Experience

Our hypothesis that longer-term physical and mental health sequelae occur as a result of PM exposure also raises further questions. Having established potential mechanisms which confer a longer-term risk of NCDs with air pollution, an important question is what exposure period is required to confer such risk? There is an association between chronic exposure to air pollution and health risks and we have also discussed behavioural and neurological changes from chronic pollutant exposure. However, with recent changes in air pollution patterns being observed as a result of the COVID-19 pandemic, and as countries are increasingly working together to reduce emissions and air pollutions, the level and components of air pollution continues to evolve. Similarly, as global warming continues to change weather patterns towards more extreme events, it may be possible that increased natural disasters such as bushfires will generate much higher PM exposure that may induce a stress response which is sufficient to disrupt the PINE network.

Even transient exposure to PM is associated with an increased risk of acute myocardial infarction [208]; however, impacts on other (non-cardio-respiratory) chronic disorders are less clear. Variability can arise with regard to the specific severity of existing conditions, source of exposure, geographical location, PM size and the individuals’ overall health [209,210] described in their analysis of the acute and chronic effects of the London Fog, that acute exposure to Total Suspended Matter (TSM) did result in increased mortality and morbidity rates in the first week following the initial event, continuing over following weeks. There was a noticeable correlation between the initial event, increased hospital admissions and insurance claims. A more recent study of forest fire PM demonstrated that in the acute period after the initial event there would be a 21% increase in general respiratory physician visits [211]. While significant, this is still lower than the 163% increase documented by [210], which is likely attributable to the composition of the PM, with coal being the predominant pollutant at the time of their study.

Several studies demonstrate an increase in respiratory and cardiovascular morbidity and mortality with chronic exposure to PM and noxious gases from diesel pollution in the urban setting, with multiple mechanistic models postulated. Acute exposure to other sources of PM, such as diesel from car exhaust, has been proven to increase airway resistance [212] and inflammation [213]; however, not to the extent discussed by Bell and Lee. Due to the nature and unpredictability of acute events such as bush fires, recording the effects of acute exposure within the population can pose its own challenges. Based on what has been observations in firefighters, not all acute exposures will result in significant changes in pulmonary function within the first few hours [214]; however, inflammatory effects from acute exposure can persist from 3 months to 10 years [215].

As for the impact on cardiovascular health, average PM2.5 has been shown to increase the risk of Atrial Fibrillation (AF) by up to 14%/24 hrs with every 5.0 µg/m3 increase, with the risk being significantly higher within the first 2 hrs of exposure [216]. It has been suggested that calculating all-cause mortality associated with PM is more accurate than specific mortality [209], with 10 µg/m3 increments in PM associated with significant increases in mortality risk [216,217].

There is little data on the long-term effects of either episodic or one-off exposure to high levels of PM. Such specific high-level exposure is not only rarer than city pollution, but considerably harder to measure, as individual exposure levels can vary significantly depending on proximity to a fire even within small geographic areas. Studies investigating this type of exposure therefore focus on cohorts of fire fighters and following notable events of severe bush fires and smog events [210,211,215]. However, recent events in Australia and the USA, where smoke pollution levels have been sustained at alarmingly high levels over prolonged periods (and geographic areas), emphasise the need for further work in this area.

## 7. Wider Implications: From Environmental Stressors to Syndemics

This work, as an example, presents an examination of a single physical environmental stressor and its impact on the PINE network, with the assertion that subsequent PINE network disruption can be linked to a wide range of NCDs, with possible but as yet limited links to chronic mental illness in the case on PM. Other physical environmental stressors may have similar impact, transduced via the PINE network, including physical stressors related to climate change [218,219,220], physical crowding [221] or noise pollution [222,223,224]. If viewed in concert with psychosocial stressors, the cumulative impact of both physical and psychosocial environmental factors on the PINE network may be substantial.

Introduced in the 1990s by Singer, the syndemic model was initially applied to the interactions among substance abuse, violence and AIDS (SAVA) [225]. Investigators noted that a variety of factors influenced risk, including structural (poverty, housing) and social factors (stigma, lack of support systems), that were strongly interlinked and potentially cumulative in impact. Subsequent syndemics have been described, including the HIV-malnutrition-food insecurity syndemic in sub-Saharan Africa [226], and a violence, immigration, depression, type 2 diabetes and abuse (VIDDA) syndemic in female Mexican immigrants in the USA [227,228]. Similar factors are identified across such syndemics, including the impact of rapid social and socioeconomic change [229].

Recent substantial public health changes and economic shifts due to COVID-19 have been particularly impactful, in some cases exaggerating economic and social inequities [230] and straining systems of medical and social support. The PINE network may adopt a central position within syndemics, serving to transduce common social, economic and environmental risks to inter-related NCDs, modifying stress and disease vulnerability and trans-generationally embedding such changes in at-risk populations.

## 8. Conclusions

Although throughout the paper, we critically discuss the existing literature relating to PM exposure, physical health consequences and potential mental health outcomes utilising the PINE network as a biological model, this does not constitute a systematic review of the literature.

We both identify, and are limited by, the lack of empirical data relating to long-term mental health outcomes as a biological, rather than psychological, result of chronic and acute PM exposure. This lack of data limits the disruption to the biological sub-networks to discussion, allowing the potential outcomes and underlying mechanisms to be hypothesised, but not tested, to conclusively determine or establish a uni-directional causality.

Focused research into the long-term mental health sequelae of air pollution and PM exposure is needed. Such research may obviate the need for the long-term screening and monitoring of the mental health of populations affected by high levels of air pollution, as well as individuals affected by bush fire disasters, possibly for months or years after an event such as the early 2020 Australian bushfires.

Such studies, together with further work on the molecular to systemic impacts of PM and their roles in acute and chronic disease, are important in informing public health strategies, which are currently lacking [231] despite the ongoing rise in air pollution and increases in both bushfire frequency and intensity in coming years. Finally, we have shown that specific environmental factors such as PM can be seen as physical environmental stressors which adversely affect the PINE network. Further work should examine other possible environmental factors and their potential impact on the PINE network and therefore on population health.

## Figures and Tables

**Figure 1 ijerph-19-13226-f001:**
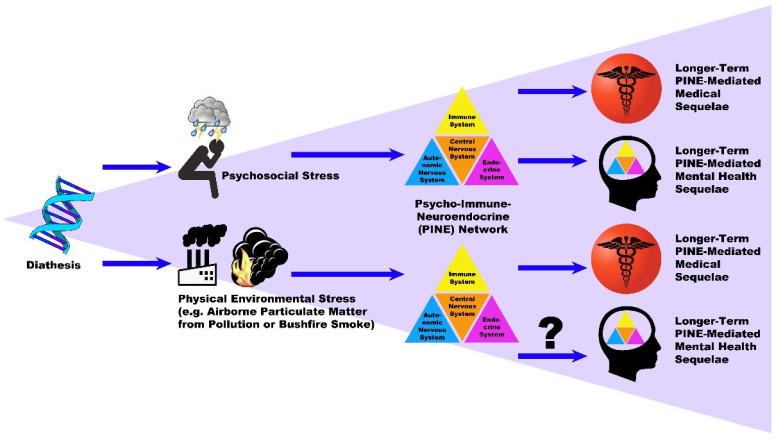
An overview of the potential relationships between diathesis, environmental stressors (e.g., PM), the PINE network, health and potential mental health sequelae.

## Data Availability

Not applicable.

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
