# Peer review of "Environmental Stressors and the PINE Network: Can Physical Environmental Stressors Drive Long-Term Physical and Mental Health Risks?"

_ijerph, 2022, doi:10.3390/ijerph192013226_

Round 1

Reviewer 1 Report

The article reviews the PINE (psycho-immune-neuroendocrine) systemic model and the effects of particulate matter on it. Being a novel topic, my suggestions to increase the attractiveness of the paper are the following:

1.- Shorten the title. For example, discard the less relevant words, such as "environmental stressors, syndemic", "physical environmental stressors", "such as", "fuel-long-term", "biologically driven". I would include the acronym PINE in the title, next to "psycho-immune-neuroendocrine". I also suggest considering the title in the form of a question.

2. The introduction and the general text seem fine. My specific suggestions are for some particular lines in the text. These are for line 171 (include more recent references), lines 239-242 (this text contributes to the overall sense of the paper, my suggestion is to discard it). Line 375 (references are quite old, it would be ideal to check for more recent literature).

Author Response

10 October 2022

Dear A/Professor Piccio, 

RE: Revisions for Manuscript ijerph-1934259

Manuscript Title: Environmental Stressors and the PINE Network: Can Physical Environmental Stressors Drive Long-Term Physcial and Mental Health Risks?

We thank you and the reviewers for your constructive feedback. Please find uploaded our revised manuscript (with track changes) by Stapelberg et al. “Environmental Stressors and the PINE Network: Can Physical Environmental Stressors Drive Long-Term Physcial and Mental Health Risks?”. Our responses to all referees are provided below. We hope you find these changes sufficient for our article to be published.

Reviewer One

The article reviews the PINE (psycho-immune-neuroendocrine) systemic model and the effects of particulate matter on it. Being a novel topic, my suggestions to increase the attractiveness of the paper are the following

1)         Shorten the title. For example, discard the less relevant words, such as "environmental stressors, syndemic", "physical environmental stressors", "such as", "fuel-long-term", "biologically driven". I would include the acronym PINE in the title, next to "psycho-immune-neuroendocrine". I also suggest considering the title in the form of a question.

We thank the reviewer for this comment. We have amended the title to make it shorter and included the acronym PINE in the new title (“Environmental Stressors and the PINE Network: Can Physical Environmental Stressors Drive Long-Term Physcial and Mental Health Risks?”). The new title now reads as a question.

2)         The introduction and the general text seem fine. My specific suggestions are for some particular lines in the text. These are for line 171 (include more recent references), lines 239-242 (this text contributes to the overall sense of the paper, my suggestion is to discard it). Line 375 (references are quite old, it would be ideal to check for more recent literature).

As suggested, we have updated the references in line 171 with the addition of Jankowska-Kieltyka et al., 2021 and Kim et al., 2020. We have also added new references in line 375 (Rider and Carlsten, 2019) and line 377 (Plusquin et al., 2017 and Prunicki et al., 2021). These have been added to the reference list.

We have also removed the suggested lines from the new manuscript.

Reviewer 2 Report

A timely paper discussing a potential syndemic between air pollution, physical, and mental health. This paper provides a good overview of the knowns and unknowns of the issue, providing context for current approaches and future work. The paper needs minor editorial revisions but the substance is sound. 

2:71 extra space before “We review the…”

3:97 extra space before “…McEwen and colleagues…”

5:169 insert space before “PM2.5 are associated…”

5:176 extra space and period before “Inhalation of poorly…”

5:181 extra space before “Volatile organic chemicals…”

5:187 add period after paragraph “…signalling (Wang et al., 2008)”

Check document throughout for extra or missing spaces and punctuation, and indented paragraphs. Two spaces are not needed after sentences.

7:272 Fix sentence “(Pope et al., 1999) also identified…” to read “Pope et al. (1999) also…”

9:383 typo “Carbon mMonoxide”

10:445-448 Awkward sentence and add space between “events,it”

11:470 typo “observedobservations”

This concept on 12:502-505 should be stated earlier in the paper to provide context: “The syndemic framework was…”

The final statement on climate change can be omitted. The paper lays out a solid framework for the hypothesis at-hand; no need to obfuscate it by introducing climate change.

Author Response

10 October 2022

RE: Revisions for Manuscript ijerph-1934259

Manuscript Title: Environmental Stressors and the PINE Network: Can Physical Environmental Stressors Drive Long-Term Physcial and Mental Health Risks?

We thank you and the reviewers for your constructive feedback. Please find uploaded our revised manuscript (with track changes) by Stapelberg et al. “Environmental Stressors and the PINE Network: Can Physical Environmental Stressors Drive Long-Term Physcial and Mental Health Risks?”. Our responses to all referees are provided below. We hope you find these changes sufficient for our article to be published.

Reviewer Two

A timely paper discussing a potential syndemic between air pollution, physical, and mental health. This paper provides a good overview of the knowns and unknowns of the issue, providing context for current approaches and future work. The paper needs minor editorial revisions but the substance is sound.

2:71 extra space before “We review the…”

We thank the reviewer for this detailed revision. This has been corrected.

3:97 extra space before “…McEwen and colleagues…”

Corrected

5:169 insert space before “PM2.5 are associated…”

Corrected

5:176 extra space and period before “Inhalation of poorly…”

Corrected

5:181 extra space before “Volatile organic chemicals…”

Corrected

5:187 add period after paragraph “…signalling (Wang et al., 2008)”

Corrected

Check document throughout for extra or missing spaces and punctuation, and indented paragraphs. Two spaces are not needed after sentences.

The document has been thoroughly screened for extra and missing spaces and indented paragraphs, and has been appropriately corrected

7:272 Fix sentence “(Pope et al., 1999) also identified…” to read “Pope et al. (1999) also…”

Corrected

9:383 typo “Carbon mMonoxide”

Corrected

10:445-448 Awkward sentence and add space between “events,it”

Corrected

11:470 typo “observedobservations”

Corrected

This concept on 12:502-505 should be stated earlier in the paper to provide context: “The syndemic framework was…”

As suggested these lines have been moved to the Introduction (lines 60-66) to provide greater context.

The final statement on climate change can be omitted. The paper lays out a solid framework for the hypothesis at-hand; no need to obfuscate it by introducing climate change.

This statement has been removed.